# Correlates of Hepatitis B infection in pregnant women attending antenatal clinics in Wa Municipality, Ghana

Emmanuel Anebakwo Awiah[1], Simon Aabalekuu[2]*, Frederick Dun-Dery[3], Elvis Dun-Dery, Junior[4], Fidelis Bayor[5], Martin Nyaaba Adokiya[6], Barnabas Bessing[7]

**1** Wa Municipal Health Administration, Upper West Region, Wa, Ghana, **2** Department of Public Health, Regional Health Directorate, Upper West Region, Wa, Ghana, **3** Department of Paediatrics, Cumming School of Medicine, University of Calgary, Alberta, Canada, **4** Department of Public Health, Purdue University, West Lafayette, IN, United States of America, **5** Department of Anaesthesiology, Lawra Municipal Hospital, Ghana Health Service, Wa, Ghana, **6** Department of Epidemiology and Biostatistics, School of Public Health, University for Development Studies, Tamale, Ghana, **7** World Health Organization, Ghana Country Office, Accra, Ghana

* aabalekuusimon78@gmail.com

**Data Availability Statement:** All the relevant data regarding this work are presented in the paper.

## Abstract

Despite the availability of an effective vaccine against viral hepatitis B infection, it remains prevalent, highly transmissible especially through mother-to-child, life-threatening, and a major public health challenge. A positive Hepatitis B e-Antigen (HBeAg) mother has a 90% risk of transmitting the virus to the unborn child in the perinatal period. This study sought to determine the prevalence and risk of Hepatitis B infection among pregnant women in the Wa Municipality of Ghana. A cross-sectional study employing systematic random sampling was conducted among 183 consented pregnant women who went for antenatal care in nine health facilities in the Wa Municipality. A structured validated questionnaire was used to collect information about socio-demographic and obstetric characteristics, awareness of Hepatitis B Virus (HBV) transmission and its prevention. Blood samples (3.0 mls) were collected from each participant to test for HBV serum markers using a Wondfo One Step HBV rapid immunochromatographic assay (Catalog number W003) for the Hepatitis B surface antigen (HBsAg). We conducted descriptive statistics including the prevalence and used multivariable logistic regression to determine the risk of Hepatitis B among study participants. Data was analysed using Stata/SE 15. About 20.2% of the 183 pregnant women screened tested positive for HBsAg. Generally, compared with younger pregnant women, older (> = 25) pregnant women were >9 times less likely to test positive for both chronic Hepatitis B core antibody (HBcAb) and (HBeAg) Hepatitis B infections. However, pregnant women in polygamous relationship were more likely to test positive for both (HBcAb) and (HBsAg and HBeAg) Hepatitis B infections compared with those in monogamous relationship. In a multivariable analysis, pregnant women in a polygamous relationships were about 5 times more likely to test positive for HBsAg (AOR = 4.61, 95% CI: 2.06–9.89) and HBcAb (AOR = 4.89, 95% CI:1.52–6.81) and HBeAg (AOR = 4.62, 95% CI:1.21–6.39) compared with those in a monogamous relationship. This study highlights a high HBsAg prevalence among pregnant women with those in polygamous relationship and younger age more likely to test positive.

**Funding:** The authors received no specific funding for this work.

**Competing interests:** The authors have declared that no competing interests exist.

Facility and community-based health services should emphasize the need for regular screening, education, and vaccination of pregnant women, especially those at high risk, to prevent mother-to-child transmission of viral hepatitis B.

## Introduction

Despite the availability of a highly effective vaccine against viral hepatitis B virus (HBV) infection, the disease has become more prevalent, life-threatening, and a tremendous public health challenge, especially in sub-Saharan Africa (5%) including Ghana, even though it is twice as much (10%) in Asia [1]. According to estimates, about 8% of Ghana's population has chronic Hepatitis B surface antigen (HBsAg) in 2020 [2, 3] with 6% among pregnant women [4]. Mother-to-Child transmission during delivery is one of the major routes of infection among children, resulting in the chronicity of the diseases with heightened risks of developing liver complications such as cirrhosis, hepatocellular carcinoma, and liver failure [5, 6]. In contrast, sexual contact is one of the transmission routes for HBV infection amongst high-risk adults in areas of low endemicity [7]. About 87%-90% of patients infected with HBV may either develop immunity or become chronic carriers, especially among those infected at infancy [8], leading to the risk of developing liver cirrhosis or liver cancer [9]. The most effective way to prevent HBV transmission to newborns is to identify HBV-positive pregnant women and timely initiation of approved interventions such as immunoglobin [10]. Thus, screening asymptomatic people is an important instrument in disease detection, prompt diagnosis, and intervention, particularly at an early stage [11]. To prevent mother-to-child transmission, a child born to a positive HBsAg mother should receive the Hepatitis B vaccine and the Hepatitis B immune globulin within 12 hours of birth, which provides over 95% protection against HBV infection [1, 12].

Women, especially of younger reproductive maternal age [13, 14], with chronic HBV infection remain a major source of the continued spread of the virus. As such, young maternal age is a high-risk factor for contracting HBV, aside from polygamous marital relationship [15–17], among other factors. Therefore, pregnant women need screening to detect the virus in prenatal care to enable early intervention [18].

Even though there have been studies in Ghana covering various aspects of hepatitis B among pregnant women, the evidence about the prevalence remains limited in the Upper West Region (UWR) of Ghana. A recent rural-urban study documented a low knowledge level of mother-to-child transmission among pregnant women in two districts of the UWR [19]. In 2020, a systematic review of the HBV prevalence in the Ghanaian population further emphasized the poor documentation of HBV-prevalence data across the country [20] and the lack of it in the UWR in particular [21]. Thus, this study explored the prevalence and associated factors of HBV among pregnant women in the Wa Municipality, Ghana to bridge this gap in the dearth of prevalence data.

## Materials and method

### Study design and setting

A facility-based cross-sectional study was conducted among pregnant women. A questionnaire was administered to gather key demographic data and awareness on transmission and prevention of the HBV infection in the municipality. The study was carried out in Wa Municipality, the capital town of the UWR of Ghana.

## Selection of antenatal care (ANC) health facilities

Nine public health facilities in the town of Wa offer prenatal care services, namely the UWR hospital, Wa municipal hospital, Wa urban health centre, Charia health centre, Kambali health centre, Charingu health centre, Bamahu health centre, Busa health centre, and Kpongu health center.

## Study participants and sampling

The study sample comprised pregnant women who attended antenatal care (ANC) in the Wa municipality from June 1–30, 2018. Considering the prevalence of 13.1% HBV infection among pregnant women in the country [22], the original sample size estimated was 175 based on the formula ($n = Z^2pq/d^2$) as suggested by Cochrane in 1977 [23]. However, after adjusting for a non-response rate of 5%, the estimated sample size increased to approximately 184 participants. Consecutive sampling was used for the selection of pregnant women attending the ANC. Pregnant women who met the inclusion criteria (attended antenatal services from June 1–30, 2018) were recruited to achieve the desired sample size. The proportion of study participants selected from each health facility was determined by the total number of the average registered ANC clients relative to each facility in the previous two months. The process was repeated every ANC day till the sample size was met. The regional hospital was purposively selected due to its central location and the fact that a substantial proportion of pregnant women in the municipality accessed antenatal care services from there. Pregnant women who had emergency conditions such as pre-eclampsia and eclampsia requiring urgent medical attention at the time were excluded from the study.

## Data collection

A structured questionnaire was used to collect the data. We collected data on respondents' socio-demographic characteristics, obstetric and gynaecological variables, and their awareness of mother-to-child transmission and prevention of HBV infection. We also collected three millilitres (3mls) of blood samples through venipuncture into ethylenediaminetetraacetic acid (EDTA) tube to measure the seroprevalence of HBsAg. The laboratory technologist cleaned the site with an alcohol swab before collecting three 3mls of blood for testing. We used a Wondfo One Step HBV rapid immunochromatographic assay (catalog number W003) for Hepatitis B envelop antibody (HBsAg), HBeAg, and Hepatitis B Core Antibody (HBcAb), HBeAb, and HBsAb because it is not automated and does not require any additional instrument. The Wondfo One Step rapid diagnostic test kit has relatively comparably high sensitivity ($\geq 98\%$) and specificity ($\geq 96\%$) [24] ratings as other rapid diagnostic test kits. The venous blood samples were stored under the recommended standard cold chain temperature range of 2˚C to 8˚C and transported in a vaccine carrier to Care Diagnostic Laboratory (CDL), a private laboratory in the study area. CDL is the best laboratory to produce faster and quality serum analysis results in the study area. All samples were analysed within 8 hours after collection.

**Potential risks.** This study had minimal risk of pain associated with the venipuncture procedure.

## Laboratory procedure and sample analysis

Before testing, the test kit and specimen were kept at room temperature (10˚C-30˚C). We opened the pouch at the notch to remove the cassette and placed it on a clean, flat surface. We labelled the cassette with the specimen identification number to be tested. We pipetted and filled the plastic dropper with the specimen. Holding the dropper vertically, we dispensed 1

drop of the specimen (about 40–50 μL for whole blood, 30–45 μL for serum/plasma) into the sample pad, ensuring no air bubbles. Then added one (1) drop (about 35–50 μL) of sample diluent and read the results in 15 minutes, even though some positive results were visible within a minimum of one (1) minute. We performed all the tests following the manufacturer's instructions with adequate controls.

### Reading and interpretation of results according to Medscape

Results of HBV serologic markers were reported qualitatively:

**Positive (+) test results:** (1) For HBsAg, HBsAb, HBeAg; the test was considered positive if a rose-pink band appeared and was visible in the control region and within the appropriate test region and (2) For HBeAb and HBcAb; the test was considered positive if a rose-pink band was visible only in the control region, and no colour band appeared in the appropriate test region.

**Negative (-) test results:** (1) For HBsAg, HBsAb, and HBeAg; the test was negative if a rose-pink band was visible only in the control region. No color band appears in the appropriate test region and (2) For HBeAb and HBcAb; the test was negative if the rose-pink band was visible in the control and appropriate test regions.

**Invalid test results**: the test was invalid if no visible band appeared or if there was a visible band only in the test region but not in the control region.

### Ethical considerations and consent

We obtained ethical approval from the Ghana Health Service Ethics Review Committee (GHSERC-029/01/18). We explained the study protocol, including potential risks/benefits of participation, to pregnant women in English and the local languages before their consented enrolment. Written informed consent was obtained from eligible women while verbal consent was obtained from participants who neither spoke nor understood the English language. In the case of the few minors, the research assistants contacted the parent or guarantor listed in the participant's ANC records via phone call to seek their verbal consent after the study protocol was explained to them. Questionnaires were assigned unique serial numbers without participants names or other personally identifiable information. We ensured that the serial numbers corresponded to the unique ID numbers on the sample collection containers to avoid the possibility of misclassification of samples and/test results. As the research information is purely for academic purposes, we encrypted soft copies whilst hard copies were stored under lock and key.

### Data analysis

After data collection, one participant withdrew her consent to participate. Thus, a total of 183 participants were included in the analysis. Descriptive statistics on socio-demographic, hepatitis B prevalence, and awareness variables were presented as frequencies, proportions and means using tables. A Chi-square Test was used to determine if there was a significant relationship between the covariates and our outcome variables. To evaluate associations with Hepatitis B serological markers, we used logistic regressions to obtain Odds Ratio (OR) along with 95% confidence intervals (CI). To identify the factors that remained independently associated with the Hepatitis B serological markers (our outcomes), we fitted a single multivariable logistics regression model that, at the onset, included all covariates and potential confounders for

which a Wald test of their estimated coefficient yielded p<0.05 in the univariable logistics analyses. This resulted in estimation of Adjusted Odds Ratio (AOR) with 95% CI. The model goodness of fit was assessed based on low Akaike Information Criterion (AIC), and Hosmer-Lemeshow test. Stata/SE 15 was used for all analyses.

## Results

### Socio-demographic characteristics of participants

The average (mean) age of participants was 32 years. The majority (60.1%) of the study participants were between 25–34 years of age, married (92.1%), self-employed (43.2%), had a tertiary level of formal education (34.4%), and acknowledged Islam as their religion (73.2%) as shown in Table 1.

**Table 1. Socio-demographic characteristics of pregnant women in Wa Municipality.**

| Variable | Measurement | |
|---|---|---|
| Age group (in years) | Frequency | Percentage (%) |
| 15–24 | 49 | 26.8 |
| 25–34 | 110 | 60.1 |
| ≥35 | 24 | 13.1 |
| Marital Status | | |
| Not married | 13 | 7.1 |
| Married | 170 | 92.9 |
| Occupation | | |
| Civil servant | 43 | 23.5 |
| Unemployed | 35 | 19.1 |
| Self-employed | 79 | 43.2 |
| Student | 26 | 14.2 |
| Level of education | | |
| No formal education | 28 | 15.3 |
| Primary | 18 | 9.9 |
| Junior High | 41 | 22.4 |
| Senior High/Vocational training | 33 | 18.0 |
| Tertiary | 63 | 34.4 |
| Parity | | |
| Primiparous (1) | 62 | 33.9 |
| Multiparous (2–4) | 95 | 51.9 |
| Multiparous (>4) | 26 | 14.2 |
| Stage of pregnancy (Gestation) | | |
| First trimester | 28 | 15.3 |
| Second trimester | 72 | 39.3 |
| Third trimester | 79 | 43.2 |
| Religious affiliation | | |
| Islam | 134 | 73.2 |
| Christianity | 49 | 26.8 |
| Traditionalist/Traditionalist | 0 | 0 |
| Area of residence | | |
| Urban | 112 | 61.2 |
| Rural | 71 | 38.8 |

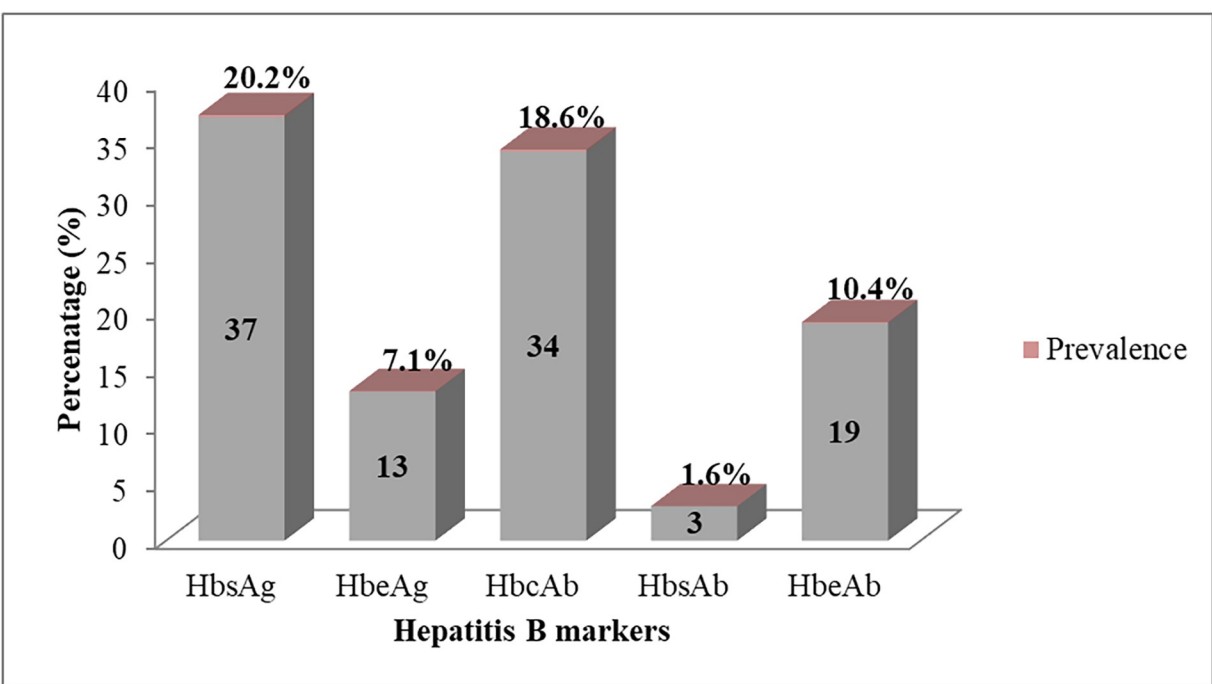

**Fig 1. The prevalence of Hepatitis B biomarkers amongst study participants.**

## Prevalence of Hepatitis B serological markers

Fig 1 shows the distribution of Hepatitis B markers among pregnant women attending ANC. Out of the 183 sera examined, 20.2%, 7.1% and 18.6% of the pregnant women tested positive for HBsAg, HBeAg, and HBcA, respectively.

## Awareness of Hepatitis B virus transmission and prevention

In this study, majority (89.1%) of the 183 pregnant women surveyed demonstrated awareness of HBV transmission indicating a high level of knowledge regarding this health concern. More than half (57.0%) of the respondents knew that HBV could be transmitted through unprotected sex, kissing (55.7%), and transfusion of infected blood and blood products (51.4%). In addition, 71.0% indicated that HBV could be cured (see Table 2).

## Factors associated with hepatitis B virus serological markers

From Table 3, we did not find a statistically significant relationship between occupation, marital status, gravida, gestation, educational level, residence, history of blood transfusion, body piercing, tattooing and family history of HBV infection and our outcomes (HBsAg, HBcAb, HBsAb, and HBeAb). However, there was a significant test of trend amongst age group for HBeAg (p = 0.01) and marital type for HBsAg (p = 0.01), HBcAb (p = 0.01) and HBeAb (p = 0.04).

In our univariate analyses (Table 4), occupation, marital status, gravida, gestation, educational level, residence, history of blood transfusion, body piercing, tattooing and family history of Hepatitis B, residence are not significantly associated with a positive test for HBsAg, HBcAb, HBsAb, and HBeAb. However, age and marital type are significantly associated with a

**Table 2. Awareness of Hepatitis B among pregnant women in the Wa Municipality.**

| Awareness questions | Response frequency N (%) | | |
|---|---|---|---|
| | Yes | No | Do not know |
| Ever heard of Hepatitis B | 163 (89.1) | 20 (10.9) | 0(0.0) |
| Hepatitis B can be transmitted through unsterilized needles, blades, and other sharp materials | 65 (35.5) | 118(64.5) | 0(0.0) |
| Hepatitis B can be transmitted by infected blood and blood products | 94 (51.4) | 89 (48.6) | 0(0.0) |
| Hepatitis B can be transmitted through kissing | 102 (55.7) | 81 (44.3) | 0(0.0) |
| Hepatitis B can affect any person | 138 (75.4) | 35 (19.1) | 10 (5.5) |
| Hepatitis B can be prevented by vaccination | 146 (79.8) | 30 (16.4) | 7(3.8) |
| Hepatitis B is transmitted through unsafe sex | 104 (56.8) | 79 (43.2) | 0(0.0) |
| Hepatitis B can be cured | 129 (70.5) | 49 (26.8) | 5(2.7) |
| Hepatitis B can be transmitted from an infected mother to an unborn child | 40 (21.9) | 143(78.1) | 0(0.0) |

positive test for HBsAg, HBeAg, and HBcAb. For instance, pregnant women aged 25–34 (AOR = 0.61, 95% CI:0.27–0.89) and ≥35 (AOR = 0.59, 95% CI:0.39–0.68) years were less likely to test positive for HBeAg compared with pregnant women age between 15–24 years.

From Table 5, our multivariable logistic regression analyses found that pregnant women aged 25–34 (AOR = 0.21, 95% CI:0.01–0.78) and ≥35 (AOR = 0.24, 95% CI:0.05–0.62) years were less likely to test positive for HBeAg compared with pregnant women age between 15–24 years. This was similar with HBcAb were women age between 25–34 (AOR = 0.42, 95% CI: 0.09–0.80) and ≥35 (AOR = 0.31, 95% CI:0.06–0.45) were less likely to test positive compared with pregnant women age between 15–24 years.

On the contrary, pregnant women in the polygamous relationship were 5 times more likely to test positive for HBsAg (AOR = 4.61, 95% CI: 2.06–9.89), HBcAb (AOR = 4.89, 95% CI:1.52–6.81), and HBeAg (AOR = 4.62, 95% CI: 1.21–6.39) compared to pregnant women in monogamous relationship.

## Discussion

### Prevalence and associated factors of hepatitis B virus serological markers

The study assessed the current prevalence of HBV and its associated factors among pregnant women in the Wa Municipality of Ghana. Our study reveals a 20.2% one-point seroprevalence of HBV infections among pregnant women. This prevalence is three times higher than the national prevalence [3] as well as other studies that reported between 9.7% and 16.7% HBsAg prevalence among pregnant women in Ghana and Cameroon [3, 6, 25]. Differences in study designs, variations in general HBsAg prevalence, varied laboratory test methods of sera, and vaccination uptake rates across the local study populations could explain the variations in observed prevalence between our study and previous research in Cameroon and Ghana [3, 6].

Our results further revealed that similar to many related studies, younger pregnant women (15–34 years), more than half (56.8%) of those in the urban area, and most of the formally educated pregnant women were positive for HBsAg compared to their rural counterparts[10, 25, 26], [27, 28]. The high HBsAg prevalence recorded among the younger pregnant mothers in this study complements the results of a related study in Germany, where persons below 30 years were 2.4% more likely to be HBsAg-positive [13], but contradicts findings of no infection among pregnant women in the Ho municipality of Ghana[20].

**Table 3. Chi-square test of association between participants' socio-demographic characteristics and their HBsAg, HBeAg, and HBcAb status.**

| Variable | HBsAg | | HBeAg | | HBcAb | | HBsAb | | HBeAb | |
|---|---|---|---|---|---|---|---|---|---|---|
| | (%) | p-value | (%) | p-value | (%) | p-value | (%) | p-value | (%) | p-value |
| **Age** | | | | | | | | | | |
| 15–24 | 32.4 | 0.26 | 53.9 | **0.01** | 32.3 | 0.20 | 33.3 | 0.99 | 31.6 | 0.77 |
| 25–34 | 48.8 | | 23.1 | | 57.5 | | 66.6 | | 57.9 | |
| ≥35 | 18.9 | | 23.1 | | 20.6 | | 0.0 | | 10.6 | |
| **Occupation** | | | | | | | | | | |
| Civil servant | 10.8 | 0.19 | 7.7 | 0.43 | 8.8 | 0.10 | 33.3 | 0.47 | 10.5 | 0.11 |
| Unemployed | 21.6 | | 23.1 | | 20.6 | | 33.3 | | 15.8 | |
| Self-employed | 48.7 | | 61.5 | | 50.0 | | 0.0 | | 42.1 | |
| Student | 18.9 | | 7.7 | | 20.6 | | 33.3 | | 31.6 | |
| **Marital Status** | | | | | | | | | | |
| Single | 13.5 | 0.09 | 23.1 | 0.06 | 14.7 | 0.07 | 0.0 | 0.89 | 5.3 | 0.88 |
| Married | 86.5 | | 76.9 | | 85.3 | | 100 | | 94.7 | |
| **Gravida** | | | | | | | | | | |
| Primigravida | 35.1 | 0.59 | 38.5 | 0.81 | 35.3 | 0.80 | 33.3 | 0.91 | 31.6 | 0.92 |
| Multigravida (2–4 children) | 46.0 | | 46.2 | | 44.1 | | 66.7 | | 57.9 | |
| Multigravida (>4 children) | 18.9 | | 15.4 | | 20.6 | | 0.0 | | 10.5 | |
| **Gestation** | | | | | | | | | | |
| 1st trimester | 16.2 | 0.52 | 23.1 | 0.12 | 14.7 | 0.74 | 33.3 | 0.46 | 26.3 | 0.48 |
| 2nd trimester | 48.7 | | 61.5 | | 47.1 | | 66.7 | | 31.6 | |
| 3rd trimester | 35.1 | | 15.4 | | 38.2 | | 0.0 | | 42.1 | |
| **Education level** | | | | | | | | | | |
| None | 21.6 | 0.69 | 15.4 | 0.38 | 20.6 | 0.80 | 0.0 | 0.66 | 10.5 | 0.71 |
| Primary | 10.8 | | 15.4 | | 11.8 | | 0.0 | | 15.8 | |
| JHS | 24.3 | | 38.5 | | 23.5 | | 33.3 | | 21.0 | |
| SHS/Vocational | 13.5 | | 15.4 | | 14.7 | | 0.0 | | 10.5 | |
| Tertiary | 29.7 | | 15.4 | | 29.4 | | 66.7 | | 42.1 | |
| Tribal marks | 48.7 | 0.27 | 53.9 | 1.00 | 47.1 | 0.25 | 33.3 | 0.41 | 42.1 | 0.17 |
| Blood transfusion | 16.2 | 0.51 | 7.7 | 1.00 | 17.7 | 0.48 | 33.3 | 0.51 | 10.5 | 0.77 |
| Tattoo | 13.5 | 0.79 | 15.4 | 0.67 | 11.8 | 1.00 | 0.0 | 0.51 | 5.3 | 0.31 |
| Multiple piercing | 43.2 | 0.85 | 53.9 | 0.39 | 47.1 | 0.56 | 66.7 | 0.37 | 42.1 | 0.96 |
| Family history | 7.7 | 0.47 | 7.7 | 0.47 | 17.7 | 0.82 | - | - | - | - |
| **Residence** | | | | | | | | | | |
| Urban | 56.8 | 0.57 | 69.2 | 0.77 | 58.8 | 0.85 | - | | - | |
| Rural | 43.2 | | 30.8 | | 41.2 | | - | | - | |
| **Marital type** | | | | | | | | | | |
| Polygamy | 51.4 | **0.01** | 46.2 | **0.04** | 41.2 | **0.01** | 33.3 | 0.82 | 41.2 | **0.05** |
| Monogamy | 37.8 | | 45.2 | | 47.1 | | 66.7 | | 58.2 | |
| Immunized against HBsAg (Yes only) | 13.5 | 0.09 | 23.1 | 1.00 | 11.8 | 0.07 | 33.3 | 0.92 | 10.5 | 0.23 |

Statistically significant at P<0.05

However, even though the HBsAg-positivity is much more prevalent among the young, formally educated, and urban pregnant women in this study, only less than one-tenth of them were seropositive for HBeAg. The low HBeAg is significant for clinical practice as it suggests that despite the high HBsAg prevalence rate, most infections were inactive and may not be virulent. Contrary to findings of a statistically significant association between low formal

**Table 4. Univariable logistic regression of covariates with Hepatitis B markers among pregnant women attending antenatal care in Wa Municipality.**

| Variable | HBsAg | HBeAg | HBcAb | HBsAb | HBeAb |
|---|---|---|---|---|---|
| | COR (95% CI) | COR (95% CI) | COR (95% CI) | COR (95% CI) | COR (95% CI) |
| **Age** | | | | | |
| 15–24 | Ref (1.00) | Ref (1.00) | Ref (1.00) | Ref (1.00) | Ref (1.00) |
| 25–34 | 1.03 (0.06–5.39) | **0.61 (0.27–0.89)** | **0.48 (0.08–0.90)** | 1.08 (0.41–1.79) | 1.08 (0.24–2.27) |
| ≥35 | 1.05 (0.62–5.71) | **0.59 (0.39–0.68)** | **0.42 (0.03–0.68)** | 1.06 (0.12–1.64) | 1.09 (0.40–2.80) |
| **Occupation** | | | | | |
| Civil servant | Ref (1.00) | Ref (1.00) | Ref (1.00) | Ref (1.00) | Ref (1.00) |
| Unemployed | 1.14 (0.98–1.31) | 0.85 (0.44–1.65) | 0.71 (0.33–1.53) | 1.10 (0.93–1.29) | 0.98 (0.72–1.34) |
| Self-employed | 1.07 (0.87–1.32) | 0.86 (0.48–1.56) | 0.68 (0.41–1.13) | 0.56 (0.24–1.28) | 1.01 (0.66–1.56) |
| Student | 0.79 (0.60–1.04) | 0.65 (0.35–1.19) | 1.03 (0.82–1.30) | 0.50 (0.22–1.09) | 1.04 (0.87–1.25) |
| **Marital Status** | | | | | |
| Single | Ref (1.00) | Ref (1.00) | Ref (1.00) | Ref (1.00) | Ref (1.00) |
| Married | 1.01 (0.91–1.13) | 0.93 (0.78–1.11) | 1.09 (0.56–2.12) | 0.75 (0.46–1.22) | 0.91 (0.86–1.19) |
| **Gravida** | | | | | |
| Primigravida | Ref (1.00) | Ref (1.00) | Ref (1.00) | Ref (1.00) | Ref (1.00) |
| Multigravida (2–4 children) | 0.93 (0.78–1.11) | 1.05 (0.70–1.56) | 1.25 (0.66–2.36) | 0.91 (0.73–1.12) | 0.98 (0.96–1.00) |
| Multigravida (>4 children) | 1.16 (0.74–1.82) | 1.01 (0.52–1.95) | 1.19 (0.98–1.45) | 1.00 (0.91–1.10) | 0.93 (0.85–1.01) |
| **Gestation** | | | | | |
| 1st trimester | Ref (1.00) | Ref (1.00) | Ref (1.00) | Ref (1.00) | Ref (1.00) |
| 2nd trimester | 0.91 (0.77–1.10) | 1.09 (0.56–2.12) | 1.07 (0.87–1.32) | 0.91 (0.86–1.19) | 1.03 (0.82–1.30) |
| 3rd trimester | 1.01 (0.91–1.13) | 0.93 (0.78–1.11) | 1.01 (0.66–1.56) | 0.75 (0.46–1.22) | 1.07 (0.98–1.17) |
| **Education level** | | | | | |
| None | Ref (1.00) | Ref (1.00) | Ref (1.00) | Ref (1.00) | Ref (1.00) |
| Primary | 1.11 (0.71–1.69) | 1.32 (0.62–2.80) | 1.01 (0.91–1.12) | 0.96 (0.63–1.47) | 1.68 (0.75–3.77) |
| JHS | 0.94 (0.77–1.10) | 0.95 (0.56–1.34) | 1.02 (0.86–1.22) | 1.35 (0.85–2.16) | 0.54 (0.15–1.95) |
| SHS/Vocational | 0.51 (0.13–1.92) | 1.10 (0.92–1.31) | 0.79 (0.39–1.59) | 1.41 (0.79–2.51) | 0.47 (0.12–1.77) |
| Tertiary | 1.07 (0.98–1.17) | 1.74 (0.49–2.01) | 1.39 (0.15–1.67) | 1.51 (0.04–2.18) | 1.14 (0.03–1.26) |
| Tribal marks | 1.16 (0.05–1.28) | 1.19 (0.08–1.30) | 0.49 (0.37–1.65) | 0.66 (0.05–1.88) | 1.09 (0.85–1.96) |
| Blood transfusion | 0.16 (0.94–1.98) | 0.97 (0.96–1.98) | 0.88 (0.82–1.95) | 1.04 (0.01–1.07) | 1.04 (0.07–1.07) |
| Tattoo | 1.02 (0.01–1.03) | 1.00 (0.99–1.01) | 0.99 (0.96–1.01) | 0.97 (0.91–1.03) | 0.93 (0.83–1.03) |
| Multiple piercing | 0.87 (0.79–1.96) | 1.06 (0.97–1.16) | 0.91 (0.01–1.92) | 1.00 (0.99–1.01) | 1.01 (0.99–1.01) |
| Family history | 1.04 (0.09–1.07) | 1.01 (0.98–1.05) | 0.95 (0.03–1.97) | 0.09 (0.05–1.97) | 1.04 (0.96–1.13) |
| **Residence** | | | | | |
| Urban | Ref (1.00) | Ref (1.00) | Ref (1.00) | Ref (1.00) | Ref (1.00) |
| Rural | 1.09 (0.70–1.68) | 0.91 (0.77–1.10) | 1.11 (0.92–1.33) | 1.01 (0.85–1.20) | 1.01 (0.91–1.13) |
| **Marital type** | | | | | |
| Polygamy | Ref (1.00) | Ref (1.00) | Ref (1.00) | Ref (1.00) | Ref (1.00) |
| Monogamy | **2.61 (1.27–4.04)** | **2.57 (1.42–5.74)** | **2.85 (1.41–6.79)** | 1.89 (0.27–2.82) | 2.74 (0.61–4.62) |
| Immunized against HBsAg (Yes only) | 1.46 (0.22–1.75) | 1.48 (0.25–1.75) | 1.75 (0.51–2.02) | 1.70 (0.46–1.96) | 1.51 (0.96–2.36) |

COR: crude odds ratio, Data in bold are P<0.05

education and high HBsAg-positivity in Brazil [29], the high HBsAg-positivity among formally educated in this study is likely only quantitative because this study has a higher proportion of formally educated respondents. The observed difference in seroprevalence could also be due to the type of testing equipment, the reagent and assay used for testing, or the endpoints considered in the respective studies. The highest HBsAg-positivity among the age group 15–24 years

**Table 5. Multivariable logistic regression of selected covariates with Hepatitis B markers among pregnant women attending antenatal care in Wa Municipality.**

| Variable | HBsAg | HBcAb | HBeAg |
|---|---|---|---|
| | AOR (95% CI) | AOR (95% CI) | AOR (95% CI) |
| **Age group** | | | |
| 15–24 | **Ref (1.00)** | **Ref (1.00)** | **Ref (1.00)** |
| 25–34 | 0.89 (0.01–1.78) | **0.42 (0.09–0.80)** | **0.21 (0.01–0.78)** |
| ≥35 | 0.91 (0.12–1.6) | **0.31 (0.06–0.45)** | **0.24 (0.05–0.62)** |
| **Marital type** | | | |
| Monogamy | **Ref (1.00)** | **Ref (1.00)** | **Ref (1.00)** |
| Polygamy | **4.61 (2.06–9.89)** | **4.89 (1.52–6.81)** | **4.62 (1.21–6.39)** |

AOR: Adjusted Odds Ratio; CI: Confidence Interval; data in bold are P<0.05

in this study also confirms findings of similar studies in Island and Taiwan [30, 31]. The findings on age group and HBeAg-positive were statistically significant and comparable to a study in Cameroon [32].

Regarding the vaccination rate against Hepatitis B, only about a third of pregnant women in this study were vaccinated before the screening, and about a tenth of the vaccinated tested positive for HBsAg without any statistical significance. Contrary to this finding, results from a similar study in Madagascar found no HBsAg-positive women even though the women had no history of Hepatitis B vaccination [2]. The type, method of storage of the vaccine, and the influence of the accuracy of the HBsAg test before vaccination could account for the different outcomes.

We also found that polygamous pregnant women were about five times more likely to be HBsAg-positive than those in monogamous marital relationships. This result confirms the findings of other studies conducted in Ethiopia, Kenya, and Nigeria [26, 32–34]. Generally, the known risk factors for HBV infection—blood transfusion, tribal marks, multi-piercings, and tattooing did not show any statistically significant association with HBsAg-positivity in this study. This finding confirms the results of a similar study in Ethiopia [34], but contradicts previous studies in Cameroon [35–37]. The observation of no association between HBV infection and the measured characteristics in our study compared to the Cameroonian studies could be due to varied HBV prevalence in the respective general populations, including variations in cultural and behavioural norms across the two study areas.

## Awareness of mother-to-child transmission of Hepatitis B and its prevention

Comparable to findings of similar studies [6, 32], over three-quarters of the pregnant women in the current research knew that HBV could be transmitted through risky lifestyles such as unprotected sexual intercourse [6, 32]. However, most pregnant women in this study did not know the possibility of mother-to-child transmission. This finding is contrary to a similar Chinese study where most pregnant women knew that their babies were highly susceptible to infection from their infected mothers, even in vitro, but did not know about transmission risks via unprotected sexual intercourse [38]. Compared to the Chinese study, this difference could be due to insufficient opportunities for health education on the Hepatitis B virus among the respondents in the current study.

## Limitations

This study was conducted in only four of the nine public health facilities in the Wa municipality due to resource constraints. As such the findings reported here could be different and generalizability should be applied with caution. We also acknowledge the detectability of HBsAg in our sample could be influenced by both the assay's sensitivity and the stage of the HBV infection. Early and very late stages of the disease might result in levels of HBsAg that are difficult to detect, and this variability is a critical consideration for interpreting test results accurately, and therefore the real-life applicability of our findings.

Additionally, we acknowledge that the study did not comprehensively explore all known relevant factors with the likelihood of influencing the natural history of HBV infectivity based on the dynamics of epidemiological principles, the public health principles of prevention and control and the framework of the modifiable risk factors of the population of interest. Therefore, the results presented here should be contextually interpreted and applied when needed.

## Conclusion

One in five of the pregnant women in the study area were HBsAg-positive. The findings of this study highlight the need for routine screening and vaccination of pregnant women, particularly those at high risk, to prevent the transmission of hepatitis B viral infection to their offspring. Facility and community-based antenatal health education should also focus on increasing maternal knowledge of the mother-to-child transmission route of HBV.

## Author Contributions

**Conceptualization:** Emmanuel Anebakwo Awiah, Elvis Dun-Dery, Junior.

**Data curation:** Fidelis Bayor.

**Formal analysis:** Emmanuel Anebakwo Awiah, Frederick Dun-Dery, Barnabas Bessing.

**Funding acquisition:** Emmanuel Anebakwo Awiah.

**Investigation:** Simon Aabalekuu.

**Methodology:** Emmanuel Anebakwo Awiah, Simon Aabalekuu, Frederick Dun-Dery, Elvis Dun-Dery, Junior, Martin Nyaaba Adokiya.

**Resources:** Emmanuel Anebakwo Awiah, Simon Aabalekuu, Frederick Dun-Dery, Elvis Dun-Dery, Junior, Fidelis Bayor, Martin Nyaaba Adokiya.

**Supervision:** Frederick Dun-Dery, Martin Nyaaba Adokiya, Barnabas Bessing.

**Writing – original draft:** Emmanuel Anebakwo Awiah, Simon Aabalekuu, Barnabas Bessing.

**Writing – review & editing:** Frederick Dun-Dery, Elvis Dun-Dery, Junior, Fidelis Bayor, Martin Nyaaba Adokiya, Barnabas Bessing.

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
