## [Decision Letter · Decision Letter 0]

10 Oct 2023

PGPH-D-23-01433

Hepatitis B viral infection and associated factors among pregnant women attending antenatal care in Wa Municipality, Ghana.

Dear Mr. Aabalekuu,

Thank you for submitting your manuscript to PLOS Global Public Health. After careful consideration, we feel that it has merit but does not fully meet PLOS Global Public Health’s publication criteria as it currently stands. Therefore, we invite you to submit a revised version of the manuscript that addresses the points raised during the review process.

EDITOR: 

The authors are kindly encouraged to consider all the recommendations made by the reviewers for revision,Attention has been called on the development of the variables, statistical analysis, and how the data is reflected in the discussion of findings,The abstract needs revision, particularly, the omissions in the reporting of the data related to the odds ration statistic and the corresponding 95% confidence intervals.

We look forward to receiving your revised manuscript.

Kind regards,

Nnodimele Onuigbo Atulomah, PhD

Academic Editor

Journal Requirements:

1. In the ethics statement in the Methods, you have specified that verbal consent was obtained. Please provide additional details regarding how this consent was documented and witnessed, and state whether this was approved by the IRB.

2. Please provide separate figure files in .tif or .eps format only and remove any figures embedded in your manuscript file. Please also ensure all files are under our size limit of 10MB.

Additional Editor Comments (if provided):

Two reviewers assessed the manuscript thoroughly and made their observations and recommendation. I agree with these observations as well. Considering how important the subject of the study is, especially with regards of providing better understanding of the dynamics involved with the problem phenomenon, I am requesting the authors to consider every recommendations made and revise as suggested.

Kindly address the issues raised about the theoretical basis of the study, the statistical analysis conducted. I am worried about the application of certain statistical analysis such as logistic regression which is the preferred tool for appropriately expressing the predictive value of the predictor variables. This has been inappropriately used in this study. Tidy the statistical analysis.

I would recommend this article: https://bmcpublichealth.biomedcentral.com/articles/10.1186/s12889-022-14723-3

Reviewers' comments:

Reviewer's Responses to Questions

**Comments to the Author**

1. Does this manuscript meet PLOS Global Public Health’s publication criteria? Is the manuscript technically sound, and do the data support the conclusions? The manuscript must describe methodologically and ethically rigorous research with conclusions that are appropriately drawn based on the data presented.

Reviewer #1: Yes

Reviewer #2: Yes

2. Has the statistical analysis been performed appropriately and rigorously?

Reviewer #1: Yes

Reviewer #2: No

3. Have the authors made all data underlying the findings in their manuscript fully available (please refer to the Data Availability Statement at the start of the manuscript PDF file)?

Reviewer #1: Yes

Reviewer #2: No

4. Is the manuscript presented in an intelligible fashion and written in standard English?

Reviewer #1: No

Reviewer #2: Yes

5. Review Comments to the Author

Reviewer #1: Congratulations to the authors on this amazing work directed towards tackling the lingering mother-to-child transmission of HBV infection in Sub-Saharan Africa. However, I have a few comments that may be helpful in fine-tuning your article.

1. Overall, there are some typos that need to be checked and corrected in the entire text. For instance, "highly transmissible" in the third line of the abstract should be followed by a comma before the word "especially".

2. The statement "A prospective systematic random cross-sectional study was conducted" in the methods section of the abstract appears to be vague. Is it a prospective systematic random study in terms of the design or the sampling approach?

It will be lucidly comprehended if it is rephrased as follows: A prospective cross-sectional study was conducted among 183 systematically and randomly selected pregnant women. Alternatively, it could be rephrased in a more canonical way by saying that "a prospective cross-sectional study employing systematic random sampling was conducted among 183 pregnant women attending..."

3. In the results section of the abstract, did you mean to say "being in a polygamous marital affair or relationship compared to monogamy?". If yes, kindly correct the current statement as it states "being in a polygamous marital compared to monogamy".

4. Is it the virus that is transmitted vertically (i.e., materno-foetally) or the surface antigen? Kindly recheck and probably correct the last statement in the conclusion section of the abstract.

5. If the statement "...especially in sub-Saharan Africa (5%) including Ghana, even though it is twice as much (10%) in Asia" is referring to HBV infection burden (i.e., prevalence), then you should attribute the figures to the regional prevalence of HBV across SSA and Asia, respectively, instead of inserting the figures without indicating what they describe.

6. There seems to be a conflicting sampling technique reported to have been used for participants' selection. At first, it was stated that a systematic random sampling approach was employed, and then again, consecutive sampling. If a systematic random sampling technique was used, was there a sampling frame? And what was the fixed sampling interval that followed the random start, if any (which wasn't mentioned, of course)? However, if it is consecutive sampling, then the statement on systematic random sampling should be corrected.

7. Regarding the sample size estimation, kindly include or reference the value of the proportion of the population attribute (i.e., p in the Cochran formula) that yielded a sample size of 183, if the confidence interval is at 95% and the margin of error is 5%. Did you in any way factor in the non-response proportion in the final sample size employed?

8. There is no statement or empirical evidence about the sensitivity and specificity of the employed Wondfo Rapid One-Step HBsAg test kit. This is very important for the validity of this study and for future studies that would want to leverage the same methods.

9. Since you do not report having a reference scale for the assessment of HBV-specific knowledge, it will be better if you do not use the terms "assessment" and "knowledge," as written in the data collection section of the methodology: "assessment of their knowledge of mother-to-child transmission and prevention of HBV." Hence, the word "awareness" should be uniformly used since it is obvious that you sought to capture their awareness of some characteristic features of the infection, such as transmission and prevention.

Furthermore, the awareness variable was only reported descriptively using frequency and proportions. Yet, in the methods section of the abstract, it was claimed that the variable was assessed using an awareness summary score. Where is the summary or the weighted aggregate score computed for the variable and the reference scale for the assessment? Therefore, the appropriate word or words should be used in place of "assessment".

10. Kindly rephrase the first statement under the "Data Analysis" section so that charts and tables will not appear as forms of descriptive statistics. They are both for data presentation.

-Which type of Chi-square test did you employ?

-What was the basis for the selection of just two variables for the multi-variable logistic model?

11. How did you ensure confidentiality regarding the participants' HBV test results and other data captured? Kindly include this in the ethical consideration section.

12. I think it is appropriate to express the last category of the respondents age as ≥35 (i.e., 35 years and above), instead of (35+)−44 in table 1. Because if the oldest respondents were aged 44 years and you wanted it to reflect in the age categorization, then you could have expressed the last age group as 35–44, similar to other categories.

-In the heading of Table 4, do you mean multiple/multivariable logistic regression instead of "multi-logistic regression"? Please check.

13. Check and correct the typos in this statement in the discussion section:"However, as recorded elsewhere, there was no significant association between respondents’ age group and their HB HBsAg-positivity status in this study, unlike reported elsewhere [21]".

14. "More than half of the pregnant women in the study area were HBsAg-positive". Really? Is 20.2% (i.e., HBsAg-positive results) more than half? Kindly correct the statement.

15. Any study limitations?

Reviewer #2: TITLE: The title would be better rendered as "Factors associated with Hepatitis B viral infection among pregnant women attending antenatal care in Wa Municipality, Ghana".

A. OBSERVATIONS: These are presented below;

Abstract: The objective of the study in line 4 of the abstract would greatly benefit from a revision and it is suggested that it should read thus; "This study sought to determine factors associated with the prevalence of Hepatitis B infection among pregnant women in Wa Municipality of Ghana."

The methods section of the abstract should read thus; "A cross-sectional survey study design was conducted among 183 consenting pregnant women in nine antenatal care facilities in Wa Municipality. A structured validated questionnaire was used to collect information about socio-demographic and obstetric characteristics, awareness of HBV transmission and its prevention scores. Blood samples (3.0 mls) were collected from each participant to test for HBV serum markers using a Wondfo One Step HBV rapid immunochromatographic assay (Catalog number W003) for the Hepatitis B surface antigen (HBsAg) profile. Data was analyzed using Stata/SE 15. Descriptive statistics were used for the socio-demographic, prevalence, and awareness variables. Chi-square and Fisher´s Exact Test were performed to determine the association between the variables.

The result section appear flawed considering that values of the 95% confidence intervals reported In lines 3-6 are inappropriately stated without the sample statistic and actually reflect not directly association but prediction computed from binary logistic regression analysis which is not mentioned as one of the statistical tools used. In actual fact, relationships are best expressed with regression analysis and the corresponding predictive values of the explanatory variables computed with binary logistic regression. These analysis are inadequate.

INTRODUCTION: The major revisions are needed for the introduction for observed typographical issues and lack of adequate conceptualization of modifiable risk factors of the at-risk population and their personal-level disposition that puts them at risk of an infection such as perception of the risks and consequences of an infection which includes transmitting the virus to their infants, attitudinal disposition towards preventing becoming infected and that prevention is far better than cure. These are salient factors to be considered in such a study. Throughout the introduction, the study did not address likely sources of associated modifiable and non-modifiable risk factors for HBV infection. In addressing these, how do you intend to establish scientific basis for these risk factors and demonstrate association as indicated by the title of the study? The contents of the study are basically what constitute risk factors for infection. No conceptual and theoretical framework or structured questions raised for the study to answer. Since these were not considered before planning the study protocol, it would mean going back to the drawing board and fresh data collection.

Methodology:

There is no evidence to suggest that the study design is a prospective study design because data collection does not resemble such hence it is recommended to stay with "facility-based cross-sectional survey design…".

Kindly be informed that Cochrane does not have a sample size computational formula, but this study is merely citing "sample size formula used in a study conducted by Cochrane in 1977" Kindly state this to reflect as expressed above which is the preferred syntax. Variables in the study appears not to be informed by any structured theoretical framework to confer validity to the instrument for data collection. The observed inclusion of knowledge in the study as a factor.

Data analysis is insufficient because only a few variables have been considered. Data analysis is inadequate and lack rigor such study deserves.

Discussions:

The discussion demonstrates clearly that the study was mainly laboratory study and was unable to uncover adequately the associated factors involved with observed prevalence of HBV infection among participants in the study. While knowledge was identified as a factor in the infection dynamics related to personal-level disposition of the at-risk population, this is not sufficient. The lack of adequately exploring fully important factors can be seen from the inability of the study to conceptualize the likely dynamics of infectivity within the epidemiological principles of the natural history of HBV infectivity, the public health principles of prevention and control and the theoretical framework of the modifiable risk factors of the population of interest. Therefore, a lot of factors have not been considered in this study. This makes the study weak and unable to provide proof of concept in regard to the objective of the study. With the conclusion stating that "More than half of the pregnant women in the study area were HBsAg-positive…" and that "Facility and community-based antenatal health education should also focus on increasing maternal knowledge of the mother-to-child transmission route of HBV" would not solve the problem because awareness alone is not sufficient to trigger the necessary caution and infection-prevention decision necessary to establish behaviour that translates to prevention of infection and transmission of the virus. The conclusion raises more questions than answers. It is not enough to test the population to determine the level of infection within the community but what may be responsible for this situation.

6. PLOS authors have the option to publish the peer review history of their article (what does this mean?). If published, this will include your full peer review and any attached files.

**Do you want your identity to be public for this peer review?** For information about this choice, including consent withdrawal, please see our Privacy Policy.

Reviewer #1: **Yes: **Ismail Bamidele Afolabi

Reviewer #2: **Yes: **Dr Bola Christie Atulomah

---

## [Editor Report · Decision Letter 1]

1 May 2024

PGPH-D-23-01433R1

Factors associated with Hepatitis B viral infection among pregnant women attending antenatal care in Wa Municipality, Ghana.

Dear Mr. Aabalekuu,

Thank you for submitting your manuscript to PLOS Global Public Health. After careful consideration, we feel that it has merit but does not fully meet PLOS Global Public Health’s publication criteria as it currently stands. Therefore, we invite you to submit a revised version of the manuscript that addresses the points raised during the review process.

Kindly consider the recommendations made for revision of aspects of the manuscript likely to have escaped your revision:

The presentation of the study design in both abstract and main manuscript. Kindly consider the explanations given for the suggested revision,The statistical analysis conducted and how the results have been presented and the rationale for revision,Importantly, the conclusions made which appears to give the impression of a cause-effect relationship for data analysis that infer predictive value.

We look forward to receiving your revised manuscript.

Kind regards,

Nnodimele Onuigbo Atulomah, PhD

Academic Editor

Journal Requirements:

Additional Editor Comments (if provided):

This is an important study that needs to be published to provide the understanding the authors of this study desire to share. The authors' concerns are well justified. However, there are certain minor but very important syntax issues to be resolved and straightened out to make the argument in the manuscript credible. Some were pointed out in the first round of the manuscript review but were not attended to. Please attend to these observed issues.

1. The title has been revised as recommended.

2. Abstract: The recommendation made by one of the reviewers to modify the methods section to read: "A cross-sectional survey study design was conducted among 183 consenting pregnant women in nine antenatal care facilities in Wa Municipality. A structured validated questionnaire was used to collect information about socio-demographic and obstetric characteristics, awareness of HBV transmission and its prevention scores. Blood samples (3.0 mL) were collected from each participant to test for HBV serum markers using a Wondfo One Step HBV rapid immunochromatographic assay (Catalog number W003) for the Hepatitis B surface antigen (HBsAg) profile. Data was analyzed using Stata/SE 15. Descriptive statistics were used for the socio-demographic, prevalence, and awareness variables. Chi-square and Fisher´s Exact Test were performed to determine the association between the variables.” was partially implemented. There remain the issue of the study design not attended to. The error of a “prospective” cross-sectional design is in error. This raises the question of What qualifies this study protocol to be considered a prospective study? Were data collected repeatedly throughout the period of the study for every participant?

The flaws appearing in the result presented in the abstract was pointed out in the first round of the review process but it appeared that no action was taken to revise the flawed observations.

There is gross misapplications of 95%CI for p-values throughout the results presented. Kindly note the data in line 34-35: “…monogamy (95%CI [.01-.03], p<0.01)…” appear to have missed the sample statistic that should have been placed before the 95%CI.

The conclusion in lines 39 - 40. do not emerge from the data. What the data suggest is that “...a high HBsAg prevalence, is likely involved with the type of marital relationship and young age.” rather than influenced because the data analysis was more of comparing the likelihood of an outcome of risk. This well expressed in line 36 - 38. The logistic regression is most appropriate in establishing predictive value when comparing explanatory variables likely to predict an outcome than its current misuse for association. To establish and characterize the nature of association would be best to apply multiple regression.

Kindly read through the manuscript for typographical errors and correct them.
---

## [Editor Report · Decision Letter 2]

14 Aug 2024

Correlates of Hepatitis B Infection in Pregnant Women Attending Antenatal Clinics in Wa Municipality, Ghana.

PGPH-D-23-01433R2

Dear Mr Aabalekuu,

We are pleased to inform you that your manuscript 'Correlates of Hepatitis B Infection in Pregnant Women Attending Antenatal Clinics in Wa Municipality, Ghana.' has been provisionally accepted for publication in PLOS Global Public Health.

Best regards,

Nnodimele Onuigbo Atulomah, PhD

Academic Editor

Congratulations for implementing the necessary corrections required.